# Plasma CoQ10 Status in Patients with Propionic Acidaemia and Possible Benefit of Treatment with Ubiquinol

**DOI:** 10.3390/antiox11081588

**Published:** 2022-08-16

**Authors:** Sinziana Stanescu, Amaya Belanger-Quintana, Borja Manuel Fernández-Felix, Pedro Ruiz-Sala, Patricia Alcaide, Francisco Arrieta, Mercedes Martínez-Pardo

**Affiliations:** 1Unidad de Enfermedades Metabólicas, Hospital Universitario Ramón y Cajal, IRYCIS, Crta de Colmenar Viejo, km 9, 100, PC 28034 Madrid, Spain; 2Unidad de Bioestadística Clínica, Instituto Ramon y Cajal de Investigación Sanitaria, Hospital Universitario Ramón y Cajal, Crta de Colmenar Viejo, km 9, 100, PC 28034 Madrid, Spain; 3Centro de Diagnóstico de Enfermedades Moleculares, Centro de Biología Molecular, Universidad Autónoma de Madrid, CIBERER, IdiPAZ, C/Francisco Tomás y Valiente, 7, PC 28049 Madrid, Spain; 4Unidad de Enfermedades Metabólicas, Hospital Universitario Ramón y Cajal, IRYCIS, CIBER-OBN, Crta de Colmenar Viejo, km 9, 100, PC 28034 Madrid, Spain; 5Unidad de Enfermedades Metabólicas, Hospital Universitario Ramón y Cajal, Crta de Colmenar Viejo, km 9, 100, PC 28034 Madrid, Spain

**Keywords:** propionic acidemia, CoQ10 deficiency, ubiquinol

## Abstract

Propionic acidaemia (PA) is an innate error of metabolism involving a deficiency in the enzyme propionyl-CoA carboxylase. Better control of acute decompensation episodes together with better treatment and monitoring have improved the prognosis of patients with this problem. However, long-term complications can arise in those in whom good metabolic control is achieved, the result of mitochondrial dysfunction caused by deficient anaplerosis, increased oxidative stress, and reduced antioxidative capacity. Coenzyme Q10 (CoQ10) is a nutritional supplement that has a notable antioxidative effect and has been shown to improve mitochondrial function. The present prospective, interventional study examines the plasma concentration of CoQ10 in patients with PA, their tolerance of such supplementation with ubiquinol, and its benefits. Seven patients with PA (aged 2.5 to 20 years, 4 males) received supplements of CoQ10 in the form of ubiquinol (10 mg/kg/day for 6 months). A total of 6/7 patients showed reduced plasma CoQ10 concentrations that normalized after supplementation with ubiquinol (*p*-value < 0.001), which was well tolerated. Urinary citrate levels markedly increased during the study (*p*-value: 0.001), together with elevation of citrate/methlycitrate ratio (*p*-value: 0.03). No other significant changes were seen in plasma or urine biomarkers of PA. PA patients showed a deficiency of plasma CoQ10, which supplementation with ubiquinol corrected. The urinary excretion of Krebs cycle intermediate citrate and the citrate/methylcitrate ratio significantly increased compared to the baseline, suggesting improvement in anaplerosis. This treatment was well tolerated and should be further investigated as a means of preventing the chronic complications associated with likely multifactorial mitochondrial dysfunction in PA.

## 1. Introduction

Propionic acidaemia (PA) is an innate error of metabolism caused by the deficiency of the mitochondrial enzyme propionyl-CoA carboxylase. Most patients begin to show signs and symptoms of the disorder while neonates, with acute metabolic decompensation episodes or in the first months of life when catabolism is increased, e.g., due to infection or prolonged fasting. These are characterized by hypo/hyperglycemia, ketosis, hyperlactacidaemia, hyperammonaemia, and even multi-organ failure [1]. Patients with PA can also develop long-term complications (even in those who have achieved good metabolic control), the pathogeny of which is not entirely understood [2]. Organs with high energy demands are those most affected, including the nervous system (encephalopathy, abnormal movement, epilepsy, psychomotor delay, involvement of the basal ganglia, and optic nerve atrophy), the heart (cardiomyopathy, arrhythmia, long QT interval), the skeletal muscles (myopathy), etc. [2]. Long-term complications independent of acute decompensation episodes can increase morbidity and mortality. Certainly, their appearance in those who have achieved good metabolic control suggests the classic “intoxication” theory explaining the appearance of these morbidities is insufficient. Recent studies have examined the diet used in PA treatment that can be associated with some chronic complications such as severe anemia or short stature [3,4].

Cellular energy loss due to mitochondrial dysfunction has gained importance in terms of explaining the physiopathology of PA. Certainly, patients with this problem share clinic and metabolic traits with those who have mitochondrial diseases [2]. For example, patients with neurological alterations involving the basal ganglia show lactate peaks (as measured by spectroscopy) similar to those observed in patients with Leigh syndrome [5]; cardiomyopathy is nearly always present in primary mitochondrial respiratory chain defects [6], while the optic nerve atrophy shares clinical and pathological traits seen in Leber hereditary optic neuropathy [7,8].

Several studies support the idea that mitochondrial dysfunction occurs in PA, with effects on different metabolic pathways [2,9] and causing profound alterations in oxidative phosphorylation [10,11,12,13,14,15]. Indeed, energy metabolism is affected at many levels in patients with PA (see Figure 1). The directly toxic nature of metabolites such as methylcitrate, and the lack of substrates for the tricarboxylic acid (TCA) cycle (e.g., oxaloacetate, citrate, and α-ketoglutarate), leads to the inhibition of anaplerotic reactions and alterations in the synthesis of NADH and FADH_2_ as electron donors to the respiratory chain [16]. Methylcitrate is a characteristic abnormal metabolite formed by the condensation of the propionyl-CoA with oxaloacetate and it has been proposed as an indirect marker of the mitochondrial propionyl-CoA pool since the latter cannot be directly determined [17]. Moreover, the citrate to methylcitrate ratio has been suggested as an important indicator to assess the disease severity [17,18].

Mitochondrial function may also be affected by an increase in oxidative stress and a reduction in antioxidative capacity. Fibroblasts from patients with PA have been reported to show increased levels of reactive oxygen species (ROS), and increased apoptosis [19], but to respond to antioxidant treatment [20]. In addition, in animal models, alterations have been identified in Complex III (ubiquinol-cytochrome C oxidoreductase) and Complex I (NADH-ubiquinone oxidoreductase), both dependent on CoQ10, as well as a depletion of mitochondrial DNA and an increase in ROS [12]. Moreover, muscle, hepatic, and cardiac biopsies from patients with PA plus cardiomyopathy are reported deficient in CoQ10 and Complexes I + III and II + III [14,15].

A number of functions have been described for CoQ10, the most important being the transport of electrons between Complexes I, II, and III in the respiratory chain (see Figure 1). However, CoQ10 is also a highly efficient liposoluble antioxidant that prevents membrane lipoprotein oxidation [21]. It regulates other antioxidants such as α-tocopherol and ascorbic acid, potentiating their effects [21,22]. CoQ10 modulates the permeability of the mitochondrial membrane, inhibiting events involved in apoptosis such as the depletion of ATP, the activation of caspase-9, the release of cytochrome C into the cytosol, the fragmentation of DNA, etc. [21]. It has also been reported to have anti-inflammatory effects via its regulation of the genes dependent on NFkB1 [23,24], as well as cardioprotective properties [25,26] and is involved in mitochondrial biogenesis [27]. Besides its implication in the mitochondria respiratory chain, CoQ is also a key component in the reactions mediated by other mitochondrial enzymes and, therefore, CoQ links the energy production to other metabolic pathways of the cell such as fatty acids or amino acids metabolism; for example, electron transfer flavoprotein dehydrogenase (ETFDH) and proline dehydrogenases (PRODH and PRODH2) use CoQ as electron acceptors [27].

The present study analyzes the plasma CoQ10 status of patients with PA and the possible benefits of supplementation with oral ubiquinol.

## 2. Materials and Methods

### 2.1. Patients and Analyses

All the patients (*n* = 7) in this prospective, interventional study had a diagnosis of PA and were monitored at the *Unidad de Enfermedades Metabólicas del Hospital Ramón y Cajal* (Madrid, Spain) (see Figure 2). All began oral supplementation with CoQ10 in the form of ubiquinol (10 mg/kg/day, divided into three doses, for 6 months). Their dietary and pharmacological treatments were not modified. Those patients already taking CoQ10 supplements were asked to stop one month before starting the study. Urine and blood analyses (urine organic acids, plasma amino acids and acylcarnitines, and plasma CoQ10, all measured after a minimum 8 h fast) were performed immediately before starting treatment, and then again at 3 and 6 months. All patients underwent an electrocardiogram (ECG) and echocardiography at the beginning and end of the study. Other explorations (fundus examination, electroencephalogram, imaging, etc.) were performed as required according to the patients’ clinical manifestations.

Plasma amino acids were determined by ion exchange chromatography (visualized with ninhydrin), plasma acylcarnitine by tandem mass spectrometry (MS/MS), urine organic acids, as their trimethylsilyl derivatives, by gas chromatography/mass spectrometry (GC/MS) following treatment with urease and liquid-liquid extraction with ethyl acetate (no oximation), and plasma CoQ10 by HPLC/MS/MS (deeming normal levels to be in the range of 0.91 ± 0.35 μmol/L). All these analyses were undertaken at the *Centro de Diagnóstico de Enfermedades Moleculares* (CEDEM), *Universidad Autónoma de Madrid* (ERNDIM-certified). All remaining analyses were undertaken at the *Hospital Ramón y Cajal*. The normal values have been made with controls that did not have related problems and both the mean and the standard deviation are according to those obtained in other works that use the same technique.

### 2.2. Statistical Analysis

We fitted a multilevel linear regression model using log transformation of the metabolic parameters as dependent continuous outcomes, while time (baseline, 3 months, and 6 months) was an independent variable. We defined a two-level model for measures (first level) within patients (second level). Thus, we considered repeated measures made for each patient. We considered it statistically significant for a *p*-value < 0.05. All analyses were performed using.Stata software version 17 (StataCorp. 2021. *Stata Statistical Software: Release 17*. StataCorp LLC., College Station, TX, USA).

### 2.3. Ethics Statement

The study was approved by the Ethics Committee of the *Hospital Ramón y Cajal*. All patients (or their guardians) provided informed consent to be included.

## 3. Results

The age of the patients at the start of the study ranged from 2.5 to 20 years (4 males); Table 1 shows their demographic and clinical characteristics. The treatment was well tolerated with no adverse effects recorded. Over the study period, three patients were admitted to the hospital: one for pancreatitis, one due to bacteriemia, and one for an episode of mild hyperammonaemia (90 μmol/L) due to dietary transgression. No new signs of cardiomyopathy, optic nerve, or other neurological problems were documented during the study period.

At baseline, 6/7 patients showed low plasma CoQ10 concentrations (see Table 2). During the study period, however, they rose significantly (*p*-value < 0.001), indicating the efficacy of the treatment in this respect (see Figure 3, Table 2). Urine Krebs cycle intermediate citrate increased significantly (*p*-value: 0.001), as did the citrate to methylcitrate ratio (*p*-value: 0.03) with ubiquinol supplementation. No changes were seen in plasma amino acids or acylcarnitine. Urine 3-OH-propionic acid and lactate remained unchanged over the study period (see Table 3; Figure 3).

## 4. Discussion

The present results suggest that plasma CoQ10 deficiency in patients with PA can be safely treated by oral supplementation with ubiquinol (10 mg/kg/day). During the study period, no patient showed any new signs of cardiomyopathy, optic nerve atrophy, or neurological disease. One patient developed mild pancreatitis that later resolved, and a single decompensation episode was recorded (mild hyperammonemia; 90 µmol/L) in another, but this was due to dietary transgression. Urine citrate levels markedly increased with an accompanying change in the citrate/methylcitrate ratio (see Figure 3). Overall, these results show that patients with PA have reduced plasma CoQ10 levels; supplementation with oral ubiquinol might help improve the TCA organic acids profile and prevent the mitochondrial dysfunction associated with deficient anaplerosis and possibly long-term health problems.

Propionyl-CoA is a very effective anaplerotic molecule, feeding the Krebs cycle via succinyl-CoA [18]. In PA patients this anaplerotic pathway is severely disturbed due to the lack of propionyl-CoA carboxylase and the consequent deficiency of succinyl-CoA (see Figure 1). In healthy individuals, the oxaloacetate (OAA) reacts with acetyl CoA to form citrate; in PA patients, the excess of propionyl-CoA sequestrates the OAA to form the methylcitric acid and consequently decreases citrate levels, further depleting the downstream intermediates of the Krebs cycle (see Figure 1) [16,17]. Thus, an increase in the methylcitrate to citrate ratio has been suggested as an important indicator of disease assessment, indirectly reflecting the excess of propionyl-CoA [28,29]. The deficient anaplerosis caused by the depletion of Krebs cycle acids such as citrate, succinyl-CoA, or alfa ketoglutarate, has been proposed as a mechanism for the chronic complication in patients with PA, which increases morbimortality with no apparent relation with the decompensation episodes [17].

Several authors have reported CoQ10-dependent alterations in mitochondrial respiratory chain Complexes I–III in animal models and in patients with PA. Indeed, alterations in energy and redox metabolism in different tissues have been recorded in different tissues from a rat PA model, with a reduction in Complex III (ubiquinol-cytochrome C oxidoreductase) activity in muscle and the brain, and in Complex I (NADH-ubiquinone oxidoreductase) activity in cardiac muscle [12]. A case has also been described in which a patient with PA, who had achieved good metabolic control, developed a rapidly fatal form of hypertrophic cardiomyopathy in which the cardiac activity of NADH cytochrome C oxidase (Complex I + III) was entirely absent [13]. Patients have also been described in whom dilated cardiomyopathy was reversed with high doses of CoQ10 (up to 25 mg/kg/day). Earlier treatment at a smaller dose (1.5 mg/kg/day) had no effect [15]. A patient with PA and dilated cardiomyopathy has also been reported with OXPHOS and CoQ10 deficiency plus significantly reduced Complex I + III and II + III activity in the liver and skeletal muscle, all of which normalized after treatment with ubiquinone [14]. Mitochondrial dysfunction in PA has also been associated with the appearance of optic nerve atrophy since it shares clinical and pathological traits with Leber hereditary optic neuropathy [7,8,30]. A similar problem has also been described in patients with methylmalonic acidaemia, but CoQ10 supplementation in such patients has returned inconsistent results [8,31,32].

Coadjutant therapy with antioxidants has been proposed as a means of optimizing mitochondrial dysfunction caused by oxidative stress and reduced antioxidative capacity in patients with PA [12,20]. Such supplementation is well tolerated, has no significant adverse effects, and could help improve antioxidative capacity in patients with certain inborn errors of metabolism especially those with low plasma ubiquinone levels [33]. In vitro, antioxidant effects can be measured directly as the capacity to eliminate ROS, but this is not so easy in vivo. The absorption, metabolism, and incorporation of antioxidants into cellular, particularly mitochondrial structures, may all influence the final result. Certainly, CoQ10 supplements can differ widely in terms of dose, presentation, and bioavailability, but ubiquinol, a reduced form of CoQ10, might be among the most useful since adequate plasma CoQ10 concentrations can be achieved. In addition, ubiquinol is known to enter cell structures [34,35,36]. In patients with primary CoQ10 deficiency, it has been shown to be superior to other supplements [35]. In mice with mitochondrial encephalopathy caused by the same deficiency, supplementation with ubiquinol returned better results than ubiquinone in terms of ameliorating histopathological lesions and Complex I–III activity, both in the brain and cardiac muscle [34].

The present work suffers from a number of limitations including the small sample size and the measurement of plasma CoQ10. The gold standard requires its measurement in skeletal muscle; techniques involving fibroblasts and lymphocytes are also available [37]. Moreover, CoQ10 synthesized in the liver is released into the circulation (inside lipid particles) and it is not certain whether plasma CoQ10 concentrations properly reflect CoQ10 activity in the tissues. However, some authors have detected reduced plasma CoQ10 concentrations in patients with metabolic or neurodegenerative disease and have suggested that such deficiencies should be prevented [38]. The small number of patients (especially pediatric patients) with inborn errors of metabolism, the lack of biomarkers for their conditions [9,28], and the need for invasive biopsies to analyze therapeutic effects [37,38] impede research into how these compounds might affect mitochondrial function. In addition, the heterogeneity of the chronic complications that appear over time [39], with no apparent connection to acute decompensation episodes, makes it even harder to gather sufficiently large samples of patients for establishing treatment criteria. The present study only lasted 6 months, precluding any statement regarding improvements or prevention lasting longer than this period.

To our knowledge, this is the first study that investigates the CoQ10 status exclusively in patients with PA and how this might be affected by supplementation with ubiquinol. Together with the increase in the CoQ10 levels, during the study, we could demonstrate an improvement in the Krebs cycle organic acids profile that, in our opinion, deserves further investigation. Along with previously reported evidence [11,12,14,15], we could speculate that chronic CoQ10 deficiency has an important part in PA physiopathology with a significant contribution to the secondary mitochondrial dysfunction detected in PA patients. The implications of CoQ10 in other metabolic pathways that link amino acids or fatty acids with the TCA cycle and energy production should also be considered [27].

## 5. Conclusions

The present patients had reduced plasma concentrations of CoQ10. Supplementation with ubiquinol at 10 mg/kg/day, which was well tolerated, corrected these concentrations and also significantly increased urine citrate levels together with the improvement of citrate to methylcitrate ratio, considered a marker of disease severity. Ubiquinol should be investigated further as a means of preventing chronic complications associated with likely deficient anaplerosis and secondary mitochondrial dysfunction in patients with PA.

## Figures and Tables

**Figure 1 antioxidants-11-01588-f001:**
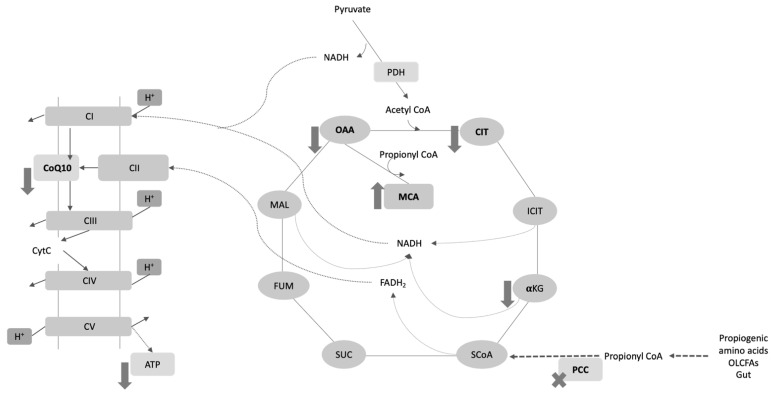
Proposed mechanisms for the mitochondrial dysfunction associated with propionic acidaemia. Methylcitrate is formed by the condensation of the propionylCoA with oxaloacetate. The lack of substrates for the tricarboxylic acid cycle (e.g., oxaloacetate, citrate and α-ketoglutarate), leads to deficient anaplerosis and alterations in the synthesis of NADH and FADH_2_ as electron donors to the respiratory chain, that are related to mitochondrial dysfunction. OAA: oxaloacetate; MCA: methylcitrate; CIT: citrate; ICIT: isocitrate; αKG: αketoglutarate, SCoA: succinyl-CoA; SUC: succinate; FUM: fumarate; MAL: malate; PCC: propionyl-CoA carboxylase; PDH: pyruvate dehydrogenase; CS: citrate synthase; CytC: cytochrome C; CI-V: mitochondrial respiratory chain complexes I-V; OLCFAs: odd long-chain fatty acids; NADH: a reduced form of nicotinamide adenine dinucleotide; FADH2: a reduced form of flavin adenine dinucleotide.

**Figure 2 antioxidants-11-01588-f002:**
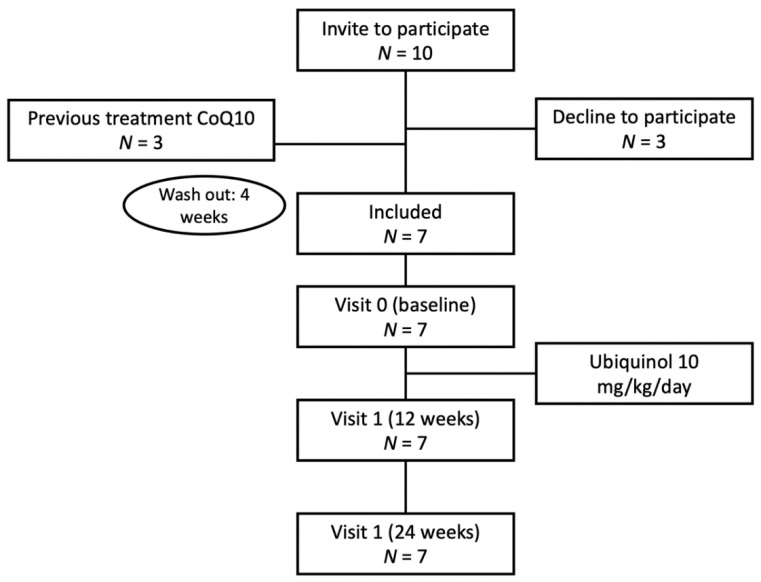
Study flow chart.

**Figure 3 antioxidants-11-01588-f003:**
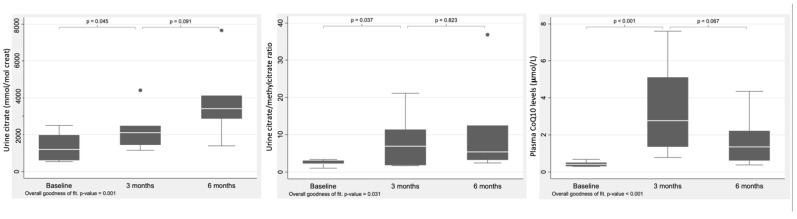
Effect of ubiquinol supplementation on plasma CoQ10 (µmol/L), urinary citrate levels (mmol/mol creat), and citrate/methylcitrate ratio.

**Table 1 antioxidants-11-01588-t001:** Patient demographic and clinical characteristics.

Sex	Age at the Beginning of the Study	Age at Diagnosis	Genetics	Clinical Course during the Study	Clinical Course (Long-Term Complications)
M	7	neonatal	PCCB gene p.Gly407Argfs*14/p.Arg410Trp	-	Severe neuromotor delay, leukopenia Severe persistent anemia
M	3	neonatal	PCCA gene p.Leu470Arg/p.Leu470Arg	Bacteriemia related to vascular central catheter	Severe neuromotor delay Choreoathetosis, basal ganglia involvement, leukopenia, frequent infections, dilated cardiomyopathy, pancreatitis Severe persistent anemia
F	10	4 months	PCCA gene p.Gly477fs*9/p.Cys616_Val633del	Mild pancreatitis	Pancreatitis
M	5	Neonatal screening	PCCB p.Asn536Asp/p.Asn536Asp	-	Autism
F	22	4 months	PCCB gene p.Gly407Argfs*14/p.Glu168Lys	-	-
M	7	6 months	PCCA gene p.Gly477fs*9/p.Cys616_Val633del	-	Sever neuromotor delay
F	13	6 months	PCCB gene p.Arg512Cys/p.Gly255Ser	Mild hyperammonemia due to dietary transgression	Neuromotor delay, epilepsy, pancreatitis, myositis

**Table 2 antioxidants-11-01588-t002:** A total of 6/7 patients presented low levels of plasma CoQ10 at the beginning of the study, which was corrected by treatment with ubiquinol.

		Plasma CoQ10 (NV: 0.91 ± 0.35 μmol/L)						
	Time	1	2	3	4	5	6	7
Patient								
Baseline		0.43	0.68	0.51	0.3	0.31	0.41	0.41
3 months		7.6	1.38	1.36	4.29	5.11	0.78	2.77
6 months		4.35	0.85	0.62	0.38		1.86	2.22

**Table 3 antioxidants-11-01588-t003:** Changes in relevant analytes over the study period. The bold characters evidence the significant variations.

	*N* Median (25th Percentile; 75th Percentile)		
	Baseline	3 Months	6 Months
CoQ10 (μmol/L) NV: 0.91 ± 0.35	*N* = 7 0.41 (0.36; 0.47)	*N* = 7 2.77 (1.37; 4.7)	*N* = 6 1.35 (0.67; 2.13)
Glutamine (μmol/L) NV: 440 ± 99	*N* = 7 387 (320; 415)	*N* = 6 439 (339; 570)	*N* = 7 381 (327; 413)
Alanine (μmol/L) NV: 283 ± 88	*N* = 7 326 (285; 350)	*N* = 6 332 (262; 376)	*N* = 7 334 (282; 445)
Lysine (μmol/L) NV: 132 ± 32	*N* = 7 118 (101.5; 155)	*N* = 6 162 (143; 179)	*N* = 7 144 (124.5; 159)
Glycine (μmol/L) NV: 209 ± 66	*N* = 7 744 (353.5; 925.5)	*N* = 6 525 (280; 1061.5)	*N* = 7 746 (404; 859)
Lactate (mmol/mol creat) NV: 1–113	*N* = 6 31.5 (21.5; 47)	*N* = 6 25 (15; 30.5)	*N* = 7 31 (23; 89)
3-OH-Propionic acid (mmol/mol creat) NV: 5–27	*N* = 6 168.5 (87; 646)	*N* = 6 181.5 (114; 396)	*N* = 7 337 (210; 578.5)
α-Ketoglutarate (mmol/mol creat) NV: 17–492	*N* = 6 35.5 (32; 186)	*N* = 6 60.5 (45.5; 220)	*N* = 7 90 (60.5; 137.5)
Citrate (mmol/mol creat) NV: 205–1735	*N* = 6 1204.5 (739; 1809)	*N* = 6 2110.5 (1522.5; 2479.5)	*N* = 7 3412 (3042; 3859)
Methylcitrate (mmol/mol creat) NV: 1–13	*N* = 5 516 (509; 811)	*N* = 5 635 (127; 676)	*N* = 7 530 (270; 906.5)
Citrate/methylcitrate	*N* = 5 2.2 (2.1; 3)	*N* = 5 6.9 (1.8; 11.4)	*N* = 7 5.4 (3.8; 9.5)
Fumarate (mmol/mol creat) NV: 0–17	*N* = 6 8 (6; 50)	*N* = 6 6 (1.5; 20)	*N* = 7 25 (9; 56)
Malate (mmol/mol creat) NV: 0–35	*N* = 6 7.5 (5.2; 81)	*N* = 6 15 (8; 20)	*N* = 7 29 (11; 72)
Succinate (mmol/mol creat) NV: 3–80	*N* = 5 35 (16; 39)	*N* = 6 50 (35; 62)	*N* = 6 49 (49; 50)

## Data Availability

All data generated or analyzed during this study are included in this published article. Upon request authors will send relevant documentation or data in order to verify the validity of the results presented (Appendix A).

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
