# Peer review of "Plasma CoQ10 Status in Patients with Propionic Acidaemia and Possible Benefit of Treatment with Ubiquinol"

_antioxidants, 2022, doi:10.3390/antiox11081588_

Round 1

Reviewer 1 Report

The authors have provided adequate answers to my comments and have made relevant amendments / additions to the text.

The manuscript has been fairly improved compared to the initially submitted version.

Author Response

Minor English changes have been made. 

Reviewer 2 Report

This is an interesting study which assesses the plasma CoQ10 status in PA patient pre and post ubiquinol treatment.

I just have some questions/comments for the authors:

1. Could the restricted diet of PA patients contributed to the low baseline plasma CoQ10 levels and was there any indication of mitochondrial impairment such as elevated lactate prior to ubiquinol therapy?

2. In line 101 why is the methylcitrate: citrate and important indicator of the disease? No explanation required.

3. What was the rationale for the dosage of ubiquinol used and why was this used rather than CoQ10?

4. Some more details about the reference range for plasma CoQ10 that was used in this paper would be appropriate and was age or gender found to influence CoQ10 status?

5. In line 202 it suggests that ubiquinol may be able to ameliorate mitochondrial function in PA patients, however there was no clear measurement of mitochondrial function in this study.

6. Could the increase in TCA intermediates indicate an increase in mitochondrial biogenesis?

7. Why did the plasma CoQ10 status decrease in a number of patient at 6 months when compared to 3 months?

8. Was there a clear correlation with clinical improvement/stability and CoQ10 therapy ? If so, how do the authors think CoQ10 may be inducing these benefits and why wasn`t evidence of oxidative stress assessed in patients as this would have had an important bearing on the discussion?
